# Psychometric Properties of the Malay Version of the Internet Gaming Disorder Scale-Short Form (IGDS9-SF-M): Evidence from a Sample of Malaysian Undergraduates

**DOI:** 10.3390/ijerph18052592

**Published:** 2021-03-05

**Authors:** Shiao Ling Ling, Nik Ruzyanei Nik Jaafar, Kit-Aun Tan, Norharlina Bahar, Azlin Baharudin, Ahmad Izzat Ahmad Tajjudin

**Affiliations:** 1Department of Psychiatry and Mental Health, Hospital Kajang, Kajang 43000, Selangor, Malaysia; lingshiaoling@gmail.com; 2Department of Psychiatry, Faculty of Medicine, Universiti Kebangsaan Malaysia Medical Centre, Cheras 56000, Kuala Lumpur, Malaysia; drazlin@ppukm.ukm.edu.my; 3Malaysian Society of Internet Addiction Prevention, Serdang 43400 UPM, Selangor, Malaysia; harlinabahar@yahoo.co.uk; 4Department of Psychiatry, Faculty of Medicine and Health Sciences, Universiti Putra Malaysia, Serdang 43400 UPM, Selangor, Malaysia; 5Department of Psychiatry, Prince Court Medical Centre, Kuala Lumpur 50450, Wilayah Persekutuan Kuala Lumpur, Malaysia; 6Faculty of Medicine and Health Sciences, Universiti Sains Islam Malaysia, Nilai 71800, Negeri Sembilan, Malaysia; izzattajuddin@usim.edu.my

**Keywords:** factor structure, reliability, concurrent validity, convergent validity

## Abstract

In recent years, increased interest in Internet Gaming Disorder has led to the development of the Internet Gaming Disorder Scale-Short Form. Translation and subsequent validation of such a scale are important for cross-cultural use. The aim of the present study was to examine the psychometric properties (factor structure, reliability, and validity) of the Malay Version of the Internet Gaming Disorder Scale-Short Form using a sample of Malaysian undergraduates. The present sample included 137 Malaysian undergraduates. Participants completed a self-administered online survey containing demographic items, the Malay Version of the Internet Gaming Disorder Scale-Short Form, the Problematic Online Gaming Questionnaire-Short Form, and the Malay Version of the Internet Addiction Test. The present findings confirm the one-factor model of the Malay Version of the Internet Gaming Disorder Scale-Short Form. Good reliability, as measured by Cronbach alpha, was found for the Malay Version of the Internet Gaming Disorder Scale-Short Form. The Malay Version of the Internet Gaming Disorder Scale-Short Form has demonstrated concurrent validity by significantly correlating with the Problematic Online Gaming Questionnaire-Short Form as well as demonstrated convergent validity with respect to the Malay Version of the Internet Addiction Test. The Malay Version of the Internet Gaming Disorder Scale-Short Form is a reliable and valid tool for assessing Internet Gaming Disorder in Malaysian undergraduates. As more research is still needed to confirm the status of Internet Gaming Disorder as a formal disorder, it is hoped that the Malay Version of the Internet Gaming Disorder Scale-Short Form can facilitate future research examining antecedents and consequences of Internet Gaming Disorder in a Malaysian setting.

## 1. Introduction

Video games were introduced in the early 1970s and video gaming gradually emerged as a leisure time activity [1,2]. With the advent of the Internet, video gaming has expanded to Internet gaming [3]. For players, Internet games are fun and engaging [4]. According to the Entertainment Software Association, it was reported that 21% of the computer and video game players were under the age of 18 years and were predominantly male [5]. Problematic and compulsive play in Internet games could impair one’s ability to function in various life aspects [6,7,8]. It was found to be positively related to anger control problems [9], social anxiety [10], denial coping mechanism [11], substance use [12], and interpersonal difficulties [13] but negatively related to social support [14], satisfaction with life [15], and educational performance [16]. Problematic and compulsive play in Internet games is now included in the *Diagnostic and Statistical Manual of Mental Disorders, Fifth Edition* (DSM-5; APA) as a behavioral addiction condition—henceforth referred to as Internet Gaming Disorder (IGD)—warranting more research data and clinical experience [17,18]. It is still debatable whether IGD should be included in the DSM-5 as a formal disorder since there is no consensus about the definition of IGD [19]. Cross-cultural data concerning the reliability and validity of IGD measurement are also limited [20].

### 1.1. Internet Gaming Disorder Assessment

The prevalence rates of IGD in adolescents ranged from 1.6% in the Netherlands to 9% in Singapore [21]. Undergraduate students have the second-highest level of Internet engagement among all demographic groups in Malaysia [22]. Although, it was reported that they used the Internet for interactive asynchronous and non-interactive activities (e.g., playing games) [23]. In one local study, Internet gaming has been reported to negatively affect undergraduate students’ capability in time management [24]. Local or regional data on the prevalence of IGD in Malaysia are scarce in the absence of a psychometrically validated measurement tool. As summarized by Koronczai et al. (2011), a psychometrically validated measurement tool should be comprehensive, brief, and reliable and valid (across methods of data collection, age groups, and types of informant) to facilitate clinical assessment and to design prevention and intervention programs [25].

Increased interest in IGD in recent years has led to the development of IGD scales. It appears that some tools (e.g., the 27-item Internet Gaming Disorder Scale [26]) are found to be lengthy for large scale surveys, other tools (e.g., the Video Game Addiction Scale for Children [27], the Video Game Addiction Test [28]) do not specifically capture all the DSM-5 criteria for IGD [29], and still other tools (e.g., the Structured Clinical Interview for DSM-5 Internet Gaming Disorder [30]) contain words which test takers find them difficult to understand [31]. In response, Pontes and Griffiths (2015) developed a brief, yet comprehensive scale, that is, the Internet Gaming Disorder Scale-Short Form (IGDS9-SF) for assessing IGD [32]. The IGDS9-SF has only nine items. All items were derived from the DSM-5 to feature the nine core criteria of IGD. A score of 36 and above is indicative of IGD. Translation and subsequent validation of the scales are important for cross-cultural use. Although there is no gold standard for scale translation and validation, general procedures include back translation, bilingual techniques, expert panel, and pilot testing [33]. The IGDS9-SF has been translated to languages such as Malay [2], Polish [8], Chinese [34], Italian [35], Portuguese [36], and Turkish [37]. As far as psychometric properties are concerned, evidence for the validity and reliability of the IGDS9-SF was established in the Polish [8], Malay [2], Portuguese [36], and Turkish [37] versions. The IGDS9-SF was found to be unidimensional in the Malay [2], Chinese [34], Italian [35], Portuguese [36], and Turkish [37] versions.

### 1.2. The Present Study

A first validation of the Malay Version-IGDS9-SF was carried out in Multiplayer Online Battle Arena (MOBA) gamers, showing that the scale is reliable and valid, as well as demonstrating measurement invariance across gender [2]. The present study expands on the MOBA study in at least three ways. First, rather than studying MOBA gamers aged between 18 and 29, the present study covers the general population in an undergraduate sample that is representative of Internet users in Malaysia. Second, participants were randomly selected in the present study, whereas the MOBA study recruited a non-random sample of gamers. Last but not least, the present study uses different validation measures, allowing for the assessment of concurrent and convergent validity. Taken together, the aim of the present study was to examine the factor structure, reliability, and validity of the Malay Version of the Internet Gaming Disorder Scale-Short Form (IGDS9-SF-M) in a sample of Malaysian undergraduates. First, we expect to confirm the one-factor structure of IGDS9-SF as found in studies with other language versions [2,34,35,36,37,38]. Second, we hypothesize that the IGDS9-SF-M would have a good internal reliability, similar to earlier findings in the MOBA study and to findings in Hong Kong, Italy, Poland, and Portugal that used other language versions of the IGDS9-SF to measure IGD. Third, we hypothesize that our study confirms the criterion concurrent and convergent validity of the IGDS9-SF-M, with the scale correlating with corresponding validation measures. That is, we expect the IGDS9-SF-M to be positively correlated with a criterion measure (i.e., the Problematic Online Gaming Questionnaire-Short Form) and with an IGD-related measure (i.e., the Malay Version of the Internet Addiction Test).

## 2. Methods

### 2.1. Participants

As recommended by Costello et al. (2005), we used subject-to-item ratios of 1:10 to calculate our sample size [39]. The sample size requirement for a psychometric examination on a 9-item tool such as the IGDS9-SF would be at least 90. We decided to recruit 110 participants after taking an attrition rate of 20% into consideration. The present sample included 137 undergraduate students at UKM. With respect to gender distribution, 34.3% were male and 65.7% were female. This gender distribution is similar to national statistics on student enrollment in Malaysian universities [40]. The average age of the present sample was 21.9 years (*SD* = 1.55). The ethnicity breakdown for the sample revealed that 70.1% were Malay, 18.2% were Chinese, 6.6% were Indian, and 5.1% were endorsed Others. Only 3% were married.

### 2.2. Ethical Clearance and Procedure

The study protocol was approved by the Universiti Kebangsaan Malaysia Research and Medical Ethics Committee. Data collection took place from September to December 2019. Five participating faculties at Universiti Kebangsaan Malaysia (UKM) were randomly selected. At the faculty level, participants were randomly selected via the odd number and even number method. The batch representative would be approached for the respective batches where the online web survey would be sent to their batchmates. Only UKM undergraduates with an Internet access and gaming medium, and signed informed consent were recruited. All participants completed a self-administered online survey.

### 2.3. Measures

#### 2.3.1. The Malay Version of the Internet Gaming Disorder Scale-Short Form

In line with the DSM-5 core criteria for IGD, Pontes and Griffiths (2015) developed the 9-item unidimensional IGD9-SF for assessing potential negative effects of IGD on the individuals’ life in the past 12 months [18,32]. Participants rate items based on a 5-point Likert scale ranging from 1 (*never*) to 5 (*very often*). Possible scores range from 9 to 45. Following the World Health Organization’s (2013) and Gorecki et al.’s (2014) recommendations, we translated the IGD9-SF into Malay [41,42,43]. The translation and validation process took three stages. In Stage 1, we obtained permission from the original scale developer. Two bilingual psychiatrists individually translated the IGD9-SF from English to Malay, the national language in Malaysia. They also back translated the scale. A panel of psychiatrists reviewed the translated scale. The panel discussed and resolved all discrepancies together with the two bilingual psychiatrists. The initial version of the IGDS9-SF-M was finalized by the panel. In Stage 2, we pilot tested the initial version of the IGDS9-SF-M using an independent sample of 10 undergraduate medical students from UKM. Comments surrounding the inappropriate use of terminologies were obtained. After reviewing the comments, we made some refinement to obtain the final version. In Stage 3, we administered the final version of the IGDS9-SF-M along with other validation measures in a sample of 137 undergraduate students.

#### 2.3.2. The Online Gaming Questionnaire-Short Form

The Problematic Online Gaming Questionnaire-Short Form (POGQ-SF) is a 12-item self-report of problematic online gaming [44]. Participants rated items on a 5-point Likert Scale ranging from 1 (*never*) to 5 (*always*). In addition to a POGQ-SF total score, six subscale scores namely Preoccupation, Overuse, Immersion, Social Isolation, Interpersonal Conflicts, and Withdrawal can be obtained. For the purposes of this study, only the POGQ-SF total scale was used. The possible range of scores for the POGQ-SF total scale would be from 12 to 60. Cronbach alpha was 0.92 for the POGQ-SF in the present study.

#### 2.3.3. The Malay Version of the Internet Addiction Test

The Malay Version of the Internet Addiction Test (MVIAT) is a 20-item self-report measure of Internet addiction [45]. Participants rated items on a 5-point Likert scale ranging from 1 (*never*) to 5 (*always*). In addition to a MVIAT total score, five subscale scores namely Lack of Control, Neglect of Duty, Problematic Use, Social Relationship Disruption, and Email Primacy can be obtained [45]. For the purposes of this study, only the MVIAT total scale was used. The possible range of scores for the MVIAT total scale would be from 20 to 100. Cronbach alpha was 0.91 for the MVIAT in the present study.

#### 2.3.4. Sociodemographics

Participants were requested to provide their information regarding age, gender, ethnicity, and marital status.

### 2.4. Data Analytic Plan

Data analyses were completed with the use of Statistical Package for the Social Sciences Version 20.0 (SPSS, Chicago, IL, USA) and AMOS. Descriptive statistics were obtained for the IGDS9-SF-M, POGQ-SF, and MVIAT. We inspected skewness and kurtosis indices for normality assumptions. The assumption of normality is violated when skewness indices exceed 3 and when kurtosis indices exceed 10 [46].

We performed confirmatory factor analyses with maximum likelihood to examine the factor structure of the IGDS9-SF-M. The IGDS9-SF was consistently found to be a single factor model in previous validation studies [2,34,35,36,37,38]. As with other language versions [2,34,35,36,37,38], we expect that the one-factor model IGDS9-SF-M would present a good fit to the present sample. Multiple goodness-of-fit indexes were used to evaluate the one-factor IGDS9-SF-M model: Ratio of Chi-Square to the Degrees of Freedom (χ^2^/*df*), Comparative Fit Index (CFI), Tucker Lewis Index (TLI), Incremental Fit Index (IFI), and Root Mean Square Error of Approximation (RMSEA). A χ^2^/*df* ratio of less than 3 represents a good model fit [47]. CFI values of 0.90 or above are described as a good fit [48]. The cut-off criteria for TLI and IFI are the same as those for the CFI. RMSEA values less than 0.05 are described as good, values between 0.05 and 0.08 as acceptable, values between 0.08 and 0.1 as marginal, and values greater than 0.1 as poor [49].

We used Cronbach alpha (α) to assess internal reliability. Cronbach alpha values are described as excellent (≥0.9), good (≥0.8), acceptable (≥0.6), questionable (≥0.5), poor (≤0.5), and unacceptable (≤0.5). [50]

As the POGQ-SF represents an established scale for assessing IGD (i.e., the same construct of interest [51]), it was used to establish concurrent validity of the IGDS9-SF-M in the present study. To this end, we examined the Pearson’s *r* correlations between the IGDS9-SF-M and POGQ-SF. A high correlation exists between Internet addiction and IGD, suggesting that these two constructs would be theoretically correlated [52]. As the MVIAT is designed to measure Internet addiction, it was used to establish convergent validity of the IGDS9-SF-M in the present study. To this end, we examined the Pearson’s *r* correlations between the IGDS9-SF-M and the MVIAT.

## 3. Results

Table 1 presents descriptive statistics for the measures in the analysis. No violation of normality assumption was detected as all skewness and kurtosis indices were within the acceptable range.

### 3.1. Factor Structure

Fitting the one-factor IGDS9-SF-M model to the present sample provided a good fit to the data, χ^2^/*df* = 2.13, CFI = 0.94, TLI = 0.92, IFI = 0.94, and RMSEA = 0.09. All items loaded significantly on the IGD construct. Standardized loadings are provided in Table 2.

### 3.2. Reliability

Good internal consistency, as measured by Cronbach alpha, was found for the IGDS9-SF-M (α = 0.87). Evidence for reliability of the IGDS9-SF-M was established in the present sample.

### 3.3. Validity

Table 3 presents intercorrelations among study measures. The IGDS9-SF-M has demonstrated concurrent validity by significantly correlating with the POGQ-SF (*r* = 0.78, *p* < 0.01). The IGDS9-SF-M has also shown convergent validity with respect to the MVIAT (*r* = 0.50, *p* < 0.01).

## 4. Discussion

IGD represents an emerging research topic in Malaysia [53,54,55,56,57]. Henceforth, the main objective of this study was to translate and validate the IGDS9-SF-M for use in Malaysian undergraduates. As far as diagnostic criteria are concerned, the IGDS9-SF has been found to feature both the DSM-5 and ICD-11 criteria of IGD. The IGDS9-SF is one of the most frequently used tools in IGD studies, thanks to its promising psychometric properties [29]. Similar to its parent scale, evidence for factor structure, reliability, concurrent validity, and convergent validity of the IGDS9-SF-M was established in the present study. As indicated by multiple indices, the present CFA findings supported the one-factor structure of the IGD construct. Consistent with other language versions [2,34,35,36,37,38], the IGDS9-SF-M was found to be a single factor model. With respect to reliability, the IGDS9-SF-M has demonstrated a good internal consistency estimate (α = 0.87), which is comparable to the original version of IGDS9-SF and other language versions such as the Brazilian version (α = 0.82) [58], the Turkish version (α = 0.89) [37], the Portuguese version (*α* = 0.87) [36], the Chinese version (α = 0.90) [34], and the Polish version (α = 0.82) [8]. The IGDS9-SF-M has demonstrated concurrent validity by significantly correlating with the POGQ-SF. To establish concurrent validity, in the present study, we paired the POGQ-SF with the IGDS9-SF-M as the former is designed to measure the same construct of interest. We also paired the MVIAT with the IGDS9-SF-M to establish convergent validity. The IGDS9-SF has been consistently shown to have consistent convergent validity with the Internet Addiction Test (IAT) in past research [29]. The IGDS9-SF-M is no exception. Evidence for the convergent validity of the IGDS9-SF-M was established. The IAT is widely used to measure Internet addiction [45]. It has five factors encompassing lack of control, neglect of duty, problematic use, social relationship disruption, and email primacy [59]. It is reasonable to speculate that Internet addiction and IGD are theoretically related constructs. This speculation, however, warrants future scientific investigations.

The notion that IGD shares psychopathological properties with Internet addiction is still debatable. Now with the existence of the IGDS9-SF-M, future research can investigate whether IGD and Internet addiction share a common developmental pathway that is essential for prevention and intervention strategies targeting the local population. In view of that fact that Internet usage and video gaming is an integral part of our daily lives, more research in the field of IGD is urgently needed. With this emerging potential new disorder of IGD, it is important to have a valid and reliable tool to be used in clinical setting and research. The IGDS9-SF-M represents a promising assessment tool that can be used by researchers to investigate IGD from a local perspective, enriching the cross-cultural IGD knowledge base. The scale can also be used by clinical practitioners in daily routine for IGD screening, and treatment planning and evaluation, when appropriate.

Several study limitations should be noted. First, given the cross-sectional nature of the present study, the predictive validity of the IGDS9-SF-M could not be determined. Second, only undergraduate students from a public university were recruited, raising the issue of generalizability. Future studies should further examine the psychometric properties of the scale in samples that are larger and more demographically diverse. Third, further evidence supporting test-retest reliability for the IGDS9-SF-M is needed. Future studies could address this limitation by asking participants to complete a second IGDS9-SF-M assessment 1 or 2 weeks after the first. Last but not least, the present study did not perform the receiver operating characteristic analysis to derive the IGDSG-SF-M cutoff scores, since no gold standard tool is available to establish the presence or absence of IGD.

## 5. Conclusions

Notwithstanding these limitations, the IGDS9-SF-M was found to be a reliable and valid tool for assessing IGD in the present sample. As more research is still needed to confirm the status of IGD as a formal disorder in the DSM-5, it is hoped that the IGDS9-SF-M can facilitate future research examining antecedents and consequences of IGD in a Malaysian setting.

## Figures and Tables

**Table 1 ijerph-18-02592-t001:** Descriptive statistics.

Study Measures	*M*	*SD*	Skewness	Kurtosis
IGDS9-SF-M	7.37	5.51	0.51	−0.58
POGQ-SF	9.65	7.96	0.66	−0.45
MVIAT	22.67	11.64	0.16	−0.65

*Note.* IGDS9-SF-M: Malay Version of the Internet Gaming Disorder Scale-Short Form; POGQ-SF: Online Gaming Questionnaire-Short Form; MVIAT: Malay Version of Internet Addiction Test.

**Table 2 ijerph-18-02592-t002:** Standardized factor loadings for the one-factor IGDS9-SF-M model in the confirmatory factor analysis.

Items	Standardized Loadings
IGDS9-SF-M_1	0.68 *
IGDS9-SF-M_2	0.71 *
IGDS9-SF-M_3	0.62 *
IGDS9-SF-M_4	0.74 *
IGDS9-SF-M_5	0.79 *
IGDS9-SF-M_6	0.66 *
IGDS9-SF-M_7	0.66 *
IGDS9-SF-M_8	0.57 *
IGDS9-SF-M_9	0.60 *

* *p* < 0.001.

**Table 3 ijerph-18-02592-t003:** Intercorrelations among study measures.

Study Measures	1	2	3
IGDS9-SF-M		0.78 *	0.50 *
POGQ-SF			0.49 *
MVIAT			

*Note.* IGDS9-SF-M: Malay Version of the Internet Gaming Disorder Scale-Short Form; POGQ-SF: Online Gaming Questionnaire-Short Form; MVIAT: Malay Version of Internet Addiction Test; ** p* < 0.01.

## Data Availability

The data presented in this study are available on request from the corresponding author.

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
