# Peer review of "Psychometric Properties of the Malay Version of the Internet Gaming Disorder Scale-Short Form (IGDS9-SF-M): Evidence from a Sample of Malaysian Undergraduates"

_ijerph, 2021, doi:10.3390/ijerph18052592_

Round 1
Reviewer 1 Report
The article is generally well written, but I have a few comments:
- The concept of "Factor structure" is not needed in the title of the paper because „control of factor structure” is part of the validity.
- I sugest to write more about the strategy of construction of the scale in the introduction.
- In addition, justify briefly the expected relationship between IGDS9-SF-M and the results of other scales. It is not entirely clear why the relation with POGQ was assesed of concurent validity and relations with MVIAT was assesed for control of convergent validity.
- In the Methods section, POGQ and MVIAT scales should be more characterized.
- In the Data Analitic Plan section, please specify the type of estimator used in the CFA. Justify why do you test the one factor model.
- The RMSEA index should be also presented in the fit indices for CFA.
- In the discussion section, CFA results should be also compared with the CFA results of other adaptations.
- In limtation of study, should be writen also about the lack of control of the test-retest reliability.
Author Response
Author Response to the Review Report (Reviewer 1)
Thank you for your valuable comments. We take your concerns seriously and have tried our best to address your points in detail. We have made the following changes in response to your comments. Please note that page and line numbers may appear differently on different computers.
Response to Reviewer 1 comments:
Point 1: The concept of "Factor structure" is not needed in the title of the paper because “control of factor structure” is part of the validity.
Response 1: We have now revised the manuscript title. It reads “Psychometric Properties of the Malay Version of the Internet Gaming Disorder Scale-Short Form (IGDS9-SF-M): Evidence From a Sample of Malaysian Undergraduates.”
Point 2: I suggest to write more about the strategy of construction of the scale in the introduction.
Response 2: As suggested, we added the strategy of construction of the scale in the Introduction, it reads:
“Increased interest in IGD in recent years has led to the development of IGD scales. It appears that some tools (e.g., the 27-item Internet Gaming Disorder Scale25) are found to be lengthy for large scale surveys, other tools (e.g., the Video Game Addiction Scale for Children26, the Video Game Addiction Test27) do not specifically capture all the DSM-5 criteria for IGD28, and still other tools (e.g., the Structured Clinical Interview for DSM-5 Internet Gaming Disorder 29) contain words which test takers find them difficult to understand.30 In response, Pontes and Griffiths (2015) developed a brief, yet comprehensive scale, that is, Internet Gaming Disorder Scale-Short Form (IGDS9-SF) for assessing IGD.31 The IGDS9-SF has only 9 items. All items were derived from the DSM-5 to feature the nine core criteria of IGD. A score of 36 and above is indicative of IGD. Translation and subsequent validation of the scales are important for cross-cultural use. Although there is no gold standard for scale translation and validation, general procedures include back translation, bilingual techniques, expert panel, and pilot testing.32” (see page 2, lines 129–143)
Point 3: In addition, justify briefly the expected relationship between IGDS9-SF-M and the results of other scales. It is not entirely clear why the relation with POGQ was assessed of concurrent validity and relations with MVIAT was assessed for control of convergent validity.
Response 3: We have now added information on the use of POGQ for concurrent validity and the MVIAT for convergent validity. It reads:
“As the POGQ-SF represents an established scale for assessing IGD (i.e., the same construct of interest51), it was used to establish concurrent validity of the IGDS9-SF-M in the present study. To this end, we examined the Pearson’s r correlations between the IGDS9-SF-M and the POGQ-SF. There exists a high correlation between Internet addiction and IGD, suggesting that these two constructs would be theoretically correlated.52As the MVIAT is designed to measure Internet addiction, it was used to establish convergent validity of the IGDS9-SF-M in the present study. To this end, we examined the Pearson’s r correlations between the IGDS9-SF-M and the MVIAT.” (see page 5, lines 288–295)
Point 4: In the Methods section, POGQ and MVIAT scales should be more characterized.
Response 4: We have provided additional psychometric information on the POGQ and the MVIAT:
“The Problematic Online Gaming Questionnaire-Short Form (POGQ-SF) is a 12-item self-report of problematic online gaming.44 Participants rated items on a 5-point Likert Scale ranging from 1 (never) to 5 (always). In addition to an POGQ-SF total score, six subscales scores namely Preoccupation, Overuse, Immersion, Social Isolation, Interpersonal Conflicts, and Withdrawal can be obtained. For the purposes of this study, only the POGQ-SF total scale was used. The possible range of scores for the POGQ-SF total scale would be from 12 to 60. Cronbach alpha was .92 for the POGQ-SF in the present study.” (see page 4, lines 244–251)
“The Malay version of Internet Addiction Test (MVIAT) is a 20-items self-report measure of Internet addiction. Participants rated items on a 5-point Likert scale ranging from 1 (never) to 5 (always). In addition to a MVIAT total score, five subscales scores namely Lack of Control, Neglect of Duty, Problematic Use, Social Relationship Disruption, and Email Primacy can be obtained.45 For the purposes of this study, only the MVIAT total scale was used. The possible range of scores for the MVIAT total scale would be from 20 to 100. Cronbach alpha was .91 for the MVIAT in the present study.” (see page 4, lines 253–259)
Point 5: In the Data Analytic Plan section, please specify the type of estimator used in the CFA. Justify why do you test the one factor model.
Response 5: As suggested, we have revised the Data Analytic Plan by specifying the type estimator used in the CFA and justifying the examination of one factor model. It reads:
“We performed confirmatory factor analyses with maximum likelihood to examine the factor structure of the IGDS9-SF-M. The IGDS9-SF was consistently found to be a single factor model in previous validation studies.33-34 As with other language versions35-36, we expect that the one-factor model IGDS9-SF-M would present a good fit to the present sample. Multiple goodness-of-fit indexes were used to evaluate the one-factor IGDS9-SF-M model: ratio of chi-square to degrees of freedom (χ2/df), comparative fit index (CFI), Tucker Lewis index (TLI), incremental fit index (IFI), and root mean square error of approximation (RMSEA). A χ2/df ratio of less than 3 represents good model fit.47-48 As noted by Hu and Bentler (1999), CFI values are described as unacceptable fit (≤ .85), reasonable fit (.85 –.95), and close fit (≥ .95).44 The cut-off criteria for TLI and IFI are the same as those for the CFI. RMSEA values less than 0.05 are described as good, values between 0.05 and 0.08 as acceptable, values between 0.08 and 0.1 as marginal, and values greater than 0.1 as poor.49” (see page 4, lines 268–279)
Point 6: The RMSEA index should be also presented in the fit indices for CFA.
Response 6: As suggested by the reviewer, we have now reported RMSEA index for CFA:
“Fitting the one-factor IGDS9-SF-M model to the present sample provided a good fit to the data, χ2/df = 2.13, CFI = .94, TLI = .92, IFI = .94, and RMSEA = 0.09.” (see page 5, lines 305–306)
Point 7: In the Discussion section, CFA results should be also compared with the CFA results of other adaptations.
Response 7: We have now highlighted our CFA results along with other language versions:
“As indicated by multiple indices, the present CFA findings supported the one-factor structure of the IGD construct.33-34 Consistent with other language versions 33-34, the IGDS9-SF-M was found to be a single factor model.” (see page 6, lines 725–728)
Point 8: In limitation of study, should be written also about the lack of control of the test-retest reliability.
Response 8: As suggested by the reviewer, we have now acknowledged the lack of test-retest reliability in the present study:
“Further evidence supporting test-retest reliability for the IGDS9-SF-M is needed. Future studies could address this limitation by asking participants to complete a second IGDS9-SF-M assessment one or two weeks after the first.”(see page 7, lines 379–381)
Thank you again for all the suggestions and comments. We hope our revisions meet your approval.

Reviewer 2 Report
Dear colleagues, I hope you are well.
Thank you for giving me the opportunity of reading the work “Factor Structure, Reliability, and Validity of the Malay Version 2 of the Internet Gaming Disorder Scale-Short Form (IGDS9-SF-3 M): Evidence From a Sample of Malaysian Undergraduates”, it has been a very big pleasure to collaborate reviewing this manuscript. The topic of this paper is very interesting and it seems necessary. However, there are several questions to improve before to publish it. I would suggest some major changes:
Title:
- The title is too long. It is recommended not to exceed 18-20 words.
Abstract:
- It is advisable to put the reader in context before to explain the aim.
- IGD acronym is not described.
- The malay version of the internet gaming disorder scale-short form cannot be a keyword.
Introduction:
- The structure of the introduction is not clear. It is recommended to divide it into sections
- Moreover, too much importance is given to validation in other countries and languages.
Method:
- Please, the first paragraph of the method should be included in the procedure.
- Add a summary table with the characteristics of the sample.
- There are important differences between sexes in terms of sample size, especially if we take into account that there are more male players than women.
Results
- The analyses carried out are insufficient. Please, look for other similar jobs to find the right procedure to follow.
Discussion
- Your conclusions are very insightful. However, I consider important to expand the limitations to give an extra information to other researchers.
Conclusion
- Ok.
References
- Please, you must follow the standards proposed by MDPI. For example:
- Year of publication in bold
- Name of the journal abbreviated and in italics.
- Add DOIs when it will be possible.
Author Response
Response to Reviewer 2 report:
Thank you for your thoughtful review of our manuscript. We take your concerns seriously and have tried our best to address your points in detail. Changes made in response to your comments and suggestions have resulted in a stronger manuscript. We hope our revision meets your approval. Please note that page and line numbers may appear differently on different computers.
Point 1:
Title: The title is too long. It is recommended not to exceed 18-20 words.
Response 1: We have now revised the manuscript title. It reads “Psychometric Properties of the Malay Version of the Internet Gaming Disorder Scale-Short Form (IGDS9-SF-M): Evidence From a Sample of Malaysian Undergraduates.”
Point 2:
Abstract: It is advisable to put the reader in context before to explain the aim. IGD acronym is not described. The Malay version of the internet gaming disorder scale-short form cannot be a keyword.
Response 2: As suggested by the reviewer, we have revised the Abstract by adding a research context and by removing all acronyms and the internet gaming disorder scale-short form keyword. It now reads:
“Increased interest in Internet Gaming Disorder in recent years has led to the development of the Internet Gaming Disorder Scale-Short Form. Translation and subsequent validation of such a scale are important for cross-cultural use. The aim of the present study was to examine the psychometric properties (factor structure, reliability, and validity) of the Malay version of the Internet Gaming Disorder Scale-Short Form using a sample of Malaysian undergraduates. The present sample included 137 Malaysian undergraduates. Participants completed a self-administered online survey containing demographic items, the Malay version of the Internet Gaming Disorder Scale-Short Form, the Problematic Online Gaming Questionnaire-Short Form, and the Malay version of the Internet Addiction Test. Good reliability, as measured by Cronbach’s alpha, was found for the Malay version of the Internet Gaming Disorder Scale-Short Form. The Malay version of the Internet Gaming Disorder Scale-Short Form has demonstrated concurrent validity by significantly correlating with the Problematic Online Gaming Questionnaire-Short Form. The Malay version of the Internet Gaming Disorder Scale-Short Form has also demonstrated convergent validity with respect to the Malay version of the Internet Addiction Test. The Malay version of the Internet Gaming Disorder Scale-Short Form is a reliable and valid tool for assessing Internet Gaming Disorder in Malaysian undergraduates. As more research is still needed to confirm the status of Internet Gaming Disorder as a formal disorder, it is hoped that the Malay version of the Internet Gaming Disorder Scale-Short Form can facilitate future research examining antecedents and consequences of Internet Gaming Disorder in Malaysian setting.
Keywords: factor structure; reliability; concurrent validity; convergent validity”. (see page 1, lines 14–33)
Point 3:
Introduction: The structure of the introduction is not clear. It is recommended to divide it into sections. Moreover, too much importance is given to validation in other countries and languages.
Response 3: As suggested by the reviewer, we have created subsections (i.e., Internet Gaming Disorder Assessment and the Present Study) in the Introduction to ensure a better flow. We hope to retain the mention of other validation studies as it provides a case for cultural diversity. Nonetheless, we have rephrased the said paragraph for greater clarity:
“The IGDS9-SF has been translated to languages such as Malay 33, Chinese 34, Italian 35, Polish 7, Portuguese 36, and Turkish 37. As far as psychometric properties are concerned, evidence for the validity and reliability of the IGDS9-SF was established in the Polish7, the Malay33, the Portuguese36, and the Turkish37versions. The IGDS9-SF was found to be unidimensional in the Chinese34 and the Italian versions35.” (see page 2, lines 285–290)
Point 4:
Method: Please, the first paragraph of the method should be included in the procedure. Add a summary table with the characteristics of the sample. There are important differences between sexes in terms of sample size, especially if we take into account that there are more male players than women.
Response 4: We have now created a separate section describing ethical clearance and procedure:
“The study protocol was approved by the Universiti Kebangsaan Malaysia Research and Medical Ethics Committee. Data collection took place from September to December 2019. Five participating faculties at Universiti Kebangsaan Malaysia (UKM) were randomly selected. At the faculty level, participants were randomly selected via the odd number and even number method. The batch representative would be approached for the respective batches where the online web survey would be sent to their batchmates. Only UKM undergraduates with Internet access and gaming medium, and signed informed consent were recruited. All participants completed a self-administered online survey.” (see page 3, lines 468–475)
To avoid redundancy, we decided not to present a socio-demographic table. Readers can find such information in the Participants.
The gender distribution is similar to national statistics on student enrollment in Malaysian universities:
“With respect to gender distribution, 34.3% were male and 65.7% were female. This gender distribution is similar to national statistics on student enrolment in Malaysian universities.40” (see page 3, 461–463).
Point 5:
Results: The analyses carried out are insufficient. Please, look for other similar jobs to find the right procedure to follow.
Response 5: Following other psychometric papers, we have revised the Data Analytic Plan. It now reads:
“Data analyses were completed with the use of Statistical Package for the Social Sciences version 20.0 (SPSS, Chicago, IL, USA) and AMOS. Descriptive statistics were obtained for the IGDS9-SF-M, the POGQ-SF, and the MVIAT. We inspected skewness and kurtosis indices for normality assumptions. The assumption of normality is violated when skewness indices exceed 3 and when kurtosis indices exceed 10.46
We performed confirmatory factor analyses with maximum likelihood to examine the factor structure of the IGDS9-SF-M. The IGDS9-SF was consistently found to be a single factor model in previous validation studies.33-34 As with other language versions35-36, we expect that the one-factor model IGDS9-SF-M would present a good fit to the present sample. Multiple goodness-of-fit indexes were used to evaluate the one-factor IGDS9-SF-M model: ratio of chi-square to degrees of freedom (χ2/df), comparative fit index (CFI), Tucker Lewis index (TLI), incremental fit index (IFI), and root mean square error of approximation (RMSEA). A χ2/df ratio of less than 3 represents good model fit.47-48 As noted by Hu and Bentler (1999), CFI values are described as unacceptable fit (≤ .85), reasonable fit (.85 –.95), and close fit (≥ .95).44 The cut-off criteria for TLI and IFI are the same as those for the CFI. RMSEA values less than 0.05 are described as good, values between 0.05 and 0.08 as acceptable, values between 0.08 and 0.1 as marginal, and values greater than 0.1 as poor.49
We used Cronbach alpha (a) to assess internal reliability. As noted by George and Mallery (2003), Cronbach alpha values are described as excellent (≥ .9), good (≥ .8), acceptable (≥ .6), questionable (≥ .5), poor (≤ .5), and unacceptable (≤ .5).50
As the POGQ-SF represents an established scale for assessing IGD (i.e., the same construct of interest51), it was used to establish concurrent validity of the IGDS9-SF-M in the present study. To this end, we examined the Pearson’s r correlations between the IGDS9-SF-M and the POGQ-SF. There exists a high correlation between Internet addiction and IGD, suggesting that these two constructs would be theoretically correlated.52As the MVIAT is designed to measure Internet addiction, it was used to establish convergent validity of the IGDS9-SF-M in the present study. To this end, we examined the Pearson’s r correlations between the IGDS9-SF-M and the MVIAT.” (see page 4, lines 551–580)
Point 6:
Discussion: Your conclusions are very insightful. However, I consider important to expand the limitations to give an extra information to other researchers.
Response 6: We would like to express our gratitude to your positive response. We have expanded the limitations and offered a few directions for future research:
“Several study limitations should be noted. First, given the cross-sectional nature of the present study, the predictive validity of the IGDS9-SF-M could not be determined. Second, only undergraduate students from a public university was recruited, raising the issue of generalizability. Future studies should further examine psychometric properties of the scale in samples that are larger, more demographically diverse. Third, further evidence supporting test-retest reliability for the IGDS9-SF-M is needed. Future studies could address this limitation by asking participants to complete a second IGDS9-SF-M assessment one or two weeks after the first. Last but not least, the present study did not perform receiver operating characteristic analysis to derive IGDSG-SF-M cutoff scores as no gold standard tool is available to establish the presence or absence of IGD.” (see page6, 761–770)
Point 7:
References: Please, you must follow the standards proposed by MDPI. For example:
- Year of publication in bold
- Name of the journal abbreviated and in italics.
- Add DOIs when it will be possible.
Response 7: Corrections were made as suggested.
Once again, we thank you for insightful comments and suggestions. We sincerely hope you will find this revision acceptable.

Round 2
Reviewer 1 Report
The authors responded quite weel to my comments. However I have one more point. The authors wrote: "As noted by Hu and Bentler (1999), CFI values are described as unacceptable fit (≤ .85), reasonable fit (.85 –.95), and close fit (≥ .95)".I don't think Hu and Bentler (1999) suggest CFI value above 0.85 as reasonable fit.
Author Response
Reviewer 1 comments: The authors responded quite well to my comments. However I have one more point. The authors wrote: "As noted by Hu and Bentler (1999), CFI values are described as unacceptable fit (≤ .85), reasonable fit (.85 –.95), and close fit (≥ .95)". I don't think Hu and Bentler (1999) suggest CFI value above 0.85 as reasonable fit.
Response to Reviewer 1 comments:
We would like to express our gratitude to your positive response. As suggested by the reviewer, we have now revised the sentence for greater clarity. It reads:
“CFI values of .90 or above are described as good fit.48” (see page 4, lines 185–186)
Please note that page and line numbers may appear differently on different computers.
Once again, we thank you for insightful comments and suggestions. We sincerely hope you will find this revision acceptable.

Reviewer 2 Report
Dear authors, the changes have been important, good job. However, the references still do not adapt to the journal's standards (the journals need to be abbreviated and the DOI indicated).
A cordial greeting.
Author Response
Reviewer 2 comments:
Dear authors, the changes have been important, good job. However, the references still do not adapt to the journal's standards (the journals need to be abbreviated and the DOI indicated).
Response to Reviewer 2 comments:
We would like to express our gratitude to your positive response. As suggested by the reviewer, we have now formatted the References as per the Journal’s style.
Thank you again for all the suggestions and comments. We hope our revisions meet your approval.
